# 24 Hours on the Run—Does Boredom Matter for Ultra-Endurance Athletes’ Crises?

**DOI:** 10.3390/ijerph19116859

**Published:** 2022-06-03

**Authors:** Christian Weich, Julia Schüler, Wanja Wolff

**Affiliations:** 1Department of Sport Science, University of Konstanz, Universitaetsstrasse 10, 78464 Konstanz, Germany; julia.schueler@uni-konstanz.de (J.S.); wanja.wolff@uni-konstanz.de (W.W.); 2Educational Psychology Lab, University of Bern, Fabrikstrasse 8, 3012 Bern, Switzerland

**Keywords:** boredom, ultra-endurance running, action crisis

## Abstract

Sport and exercise can be boring. In the general population, thinking of sports as boring has been linked to exercising less. However, less is known about the role of boredom in people who participate in ultra-endurance competitions: Do these athletes also associate their sports with boredom, and does boredom pose a self-regulatory challenge that predicts if they encounter a crisis during an ultra-endurance competition? Here, we investigate these questions with a sample of *N* = 113 (*n* = 34 female) competitors of a 24 h hour running competition, aged *M* = 37.6 ± 13.8 years. In this study, *n* = 23 *very extreme* athletes competed as single starters or in a relay team of 2, and *n* = 84 *less extreme* athletes competed in relay teams of 4 or 6. Before the run, athletes completed self-report measures on sport-specific trait boredom, as well as the degree to which they expected boredom, pain, effort, and willpower to constitute self-regulatory challenges they would have to cope with. After the run, athletes reported the degree to which they actually had to deal with these self-regulatory challenges and if they had faced an action crisis during the competition. Analyses revealed that very extreme athletes displayed a significantly lower sport-specific trait boredom than less extreme athletes (*p* = 0.024, d=−0.48). With respect to self-regulatory challenges, willpower, pain, and effort were expected and reported at a much higher rate than boredom. However, only boredom was as a significant predictor of experiencing a crisis during the competition (odds ratio = 12.5, *p* = 0.02). Our results show that boredom also matters for highly active athletes. The fact that the experience of boredom—and not more prototypical competition-induced challenges, such as pain or effort—were linked to having an action crisis highlights the relevance of incorporating boredom into the preparation for a race and to the performance management during competition.

## 1. Introduction

Boredom is an “aversive state that occurs when we are not able to successfully engage attention” and being bored is accompanied by a “high degree of mental effort expended in an attempt to engage with the task” [1]. Thus, per definition, the experience of boredom is aversive, and people strive to minimize or avoid being bored [2]. However, research shows that boredom is a frequent experience in everyday life and that those who experience it regularly and intensely (i.e., score high on boredom proneness, and thus display high trait boredom) are more likely to engage in unhealthy behaviors and less likely to engage in healthy actions (for reviews, please see [3,4]). For example, one study found that higher levels of sport-specific (the relatively stable tendency to associate sports and exercise with boredom) and domain-general trait measures of boredom were related to less physical exercise in a large sample of crowd workers [5]. In addition to trait boredom, state boredom, which refers to situationally defined influences on one’s tendency to experience boredom, is a powerful motivator of behavior [6]. Being bored has for example been linked to negative behaviors, such as self-administered pain [7] or sadistic aggression [8], but also to positive responses, such as creativity [9].

Boredom appears to exert its effects due to the role it plays in how people self-regulate their behavior. Theoretical and empirical work has shown that persisting in a boring situation places substantial self-regulatory demands on the bored person [10,11]. More specifically, being bored is understood to create the urge to do something else [11,12], thereby making it harder to continue what one is currently doing [13]. In order to continue with a boring task, the bored person therefore has to employ self-regulatory control (or “willpower”). To illustrate, a runner who experiences boredom during a run will need to employ willpower to keep going *despite* the boredom-induced urge to do something else. Consistent with this, boredom can create feelings of fatigue [14] or frustration, and even impair performance [10]. Indeed, in a high-powered experimental study, Milyavskaya et al. [14] showed that performing an easy but boring task created sensations of fatigue that could even outweigh the fatigue that is experienced by performing a supposedly more challenging task that was designed to require effortful control. More relevant for the sports and exercise context, this finding was conceptually replicated in a recent study, showing that completing either a boring or a control demanding first task did not differentially affect performance in a handgrip endurance task that was completed afterwards [15]. Thus, being bored, or having to complete a control demanding task, appeared to have similar effects on performance in a subsequent effortful physical task. Recent work outside the sporting context even found that people seem to prefer physical effort over being bored [16].

In sum, a large body of research shows that trait and state boredom play a powerful role in how people self-regulate their behavior in different life domains [17,18,19]. Moreover, the magnitude of boredom’s effects sometimes even outweighs those that are induced by tasks that are designed to require willpower or effortful control. With respect to sports and exercise, recent work demonstrates that boredom matters when it comes to understanding how much people exercise [5] and how well they perform in physically demanding tasks [15]. Crucially, the aforementioned studies refer to samples drawn from the general population and investigate if boredom can potentially deter people from engaging in sports at all, and how it can alter performance in a standardized physical effort task in the lab. However, less is known about the role of boredom in people and conditions on the other side of this continuum, and whether athletes who participate in extreme endurance events also associate sports with boredom. Do athletes experience boredom during such extreme sports competitions, and does it play a role in how athletes fare during the competition? Here, we aim to address these questions in a sample of (ultra-endurance) runners.

### 1.1. Trait Boredom and Ultra-Endurance Running

Ultra-endurance running is a type of sports that, for outsiders, seems to exhibit almost ideal properties for inducing boredom. Indeed, in ultra-endurance running, athletes have to execute the same motion over and over again at a relatively low intensity. In addition, the change in external stimulation can be quite small and gradual in many ultra-endurance events. Thus, the environment can be very monotonous: athletes might run through deserts, where the lack of specific external landmarks is so pronounced that runners are in danger of getting lost, so they even have to wear tracking devices [20]. Likewise, a competition might take so long that athletes have to compete during the night, thereby running through almost complete darkness, again with little change in external stimulation. In addition, ultra-endurance events can be incredibly demanding, posing challenges that far exceed the expectations of most athletes. This can be true even for those who signed up for the race and have specifically trained for it. To illustrate, the annual Barkley Marathons, are famous for offering such demanding race settings that, in some years, none of the athletes can even make it to the finish in time. Monotony, repetition, and demands that are so high that even excellent preparation can produce rather slim odds of success all appear to be central to ultra-endurance running. All of these factors have been linked to inducing boredom [21,22].

Despite this ostensibly high potential for inducing boredom, the popularity of ultra-endurance competitions (defined as any endurance event longer than 6 h) has risen considerably within the last 25 years [23]. Thus, although endurance sports might seem particularly boring to many people [24], an increasing number of people are drawn to participate in them. It is unlikely that these athletes perceive their sports as particularly boring, because people rarely choose to engage in activities that bore them [2], and try hard to escape boring situations [25]. To ultra-endurance athletes, their sports might even offer an escape from boredom and, thereby, be a source of meaningful and valuable experiences. Consistent with this, escapism from everyday life, experiencing nature, and to make borderline experiences are stated as factors when ultra-endurance athletes are queried about what motivates them to engage in their sports [26]. While ultra-endurance athletes in general are probably not very prone to succumb to their sports’ potential for inducing boredom (i.e., repetition, monotony, and difficulty) [27], it is plausible that even within this group, differences exist in sport-specific trait boredom. Very extreme athletes compete in the longest and most demanding competitions (e.g., races as described above), and it is conceivable that they display even lower levels of sport-specific trait boredom than their less extreme ultra-endurance peers (who do not compete in the longest competitions but rather compete in races with shorter running bouts).

### 1.2. Self-Regulatory Challenges and Action Crises during Competition: The Role of State Boredom

Does boredom matter as a self-regulatory challenge during ultra-endurance competitions? Clearly, ultra-endurance sports pose a multitude of self-regulatory challenges that likely far outweigh boredom in their magnitude and salience. For example, exercise-induced pain, as well as having to employ willpower and effort toward reaching one’s sporting goal, are frequently associated with (ultra-)endurance sports [28,29,30,31]. Indeed, subjective pain, perception of exertion, and willpower have all been described as being particularly relevant in ultra-endurance sports [32,33,34,35]. Given their relevance as determinants for success in endurance sports, successful athletes are potentially very well versed in dealing with them [36,37,38]. This can be illustrated using the example of pain. Being able to sustain very high levels of pain is critical to success in ultra-endurance sports, as athletes have to keep going even if their bodies are hurting [39]. Interestingly, ultra-endurance athletes seem to be very good at dealing with pain: A study with ultra-endurance runners who competed in a 4487 km foot race that lasted 64 days reported that the ultra-endurance athletes displayed a higher pain tolerance in a cold pressor test than controls [33]. Thus, while pain is a key self-regulatory challenge ultra-endurance athletes have to deal with, they are also very capable of coping with it [40].

If self-regulatory challenges, such as having to deal with pain, cannot be managed, it is likely that the athlete will begin to struggle until, in the worst case, he or she will have to withdraw from the competition [41]. Such a dropout, or the disengagement from the original goal (e.g., to cross the finish line), does not function like an on/off switch that is flipped abruptly. Thus, the final accomplishment of the goal is accompanied by the constant process, which keeps the athlete between the further pursuit of the desired outcome or giving up. This is defined as an *action crisis*, usually occurring when an athlete has already invested a tremendous effort in achieving her goal, but is repeatedly confronted with mental difficulties [42]. One way to assess the impact self-regulatory challenges have on success during an ultra-endurance competition is to evaluate if they predict an action crisis during competition.

The relevance of the self-regulatory challenge of dealing with pain, exertion tolerance and applying willpower is well established [32,33,34,35]. Much less is known about the role of boredom during ultra-endurance competitions. Although ultra-endurance athletes are unlikely to be prone to associate their sports with boredom (i.e., score low on sport-specific trait boredom), it is still plausible that even those athletes can get bored during a competition, and that they might encounter a crisis during their competition because of boredom. This is especially true when the duration of the competition lasts many hours, as it is typical in ultra-endurance sports. The latter is particularly critical because only a few studies have addressed boredom in the context of sports, especially in high-performance sports. As one notable exception, Hoffman and Krouse [27] reported a generally low boredom susceptibility among ultra-marathon participants. Thus, consistent with the reasoning presented here, ultra-endurance athletes seemed to be less prone to becoming bored. However, this did not address if even low levels of boredom might lead to a crisis during a competition. Highlighting the negative effect boredom can have on performance, Velasco and Jorda [43] reported that boredom was frequently experienced and did have a negative effect on performance among elite and sub-elite athletes. This was particularly the case when training consisted of repetitive tasks. Thus, consistent with the reason presented here, boredom appears to matter as a self-regulatory challenge in sports. However, this study was not conducted with (ultra-)endurance athletes who might be particularly robust against the experience of boredom [27].

### 1.3. The Present Study—24 Hours on the Run

Research on boredom among ultra-endurance athletes is lacking. Here, we tackle this gap with two research questions. First, we expect that ultra-endurance (*very extreme*) runners display lower levels of sport-specific trait boredom than their *less extreme* peers. To investigate this question, we adapted a measure of sport-specific trait boredom [5,44]. Runners were classified as *very extreme* if they competed in a 24 h running competition as single starters or duo starters. They were compared to *less extreme* runners, who participated in the same competition in relay teams of four or six competitors. This classification is based on the expected run time of at least 12 h or longer for the ultra-endurance runners, which requires a much higher endurance performance, creating even more advanced and challenging conditions.

Secondly, we assess if runners experience boredom during an ultra-running competition and, if so, whether this leads to an action crisis during their race. To address this question, the runners were surveyed before and after partaking in the 24 h running competition. To assess boredom, runners replied to a single item assessment of the degree to which they expected to get bored during the run (measured before the run), as well as the degree to which they actually experienced boredom during the run (measured after the run). To contrast boredom with more prototypical challenges, athletes also replied to single item measures on effort, willpower, and pain in the same before (*expectation*) and after (*actual experience*) format. In order to assess if any of the surveyed challenges would predict the occurrence of a crisis, athletes were queried after the race if they had encountered a crisis during their run.

## 2. Methods

### 2.1. Participants

The study included a total of *N* = 113 highly active and experienced athletes (female: *n* = 34, male: *n* = 74, did not provide answer: *n* = 7) who were between 18 and 75 years old (*M* = 37.6 years ± 13.8), and who participated in the survey before and after the 24 h race on the 12 and 13 June 2021 in Reichenau, Germany. During the race, *n* = 21 of them took part as a single starter and *n* = 2 in a duo. Those athletes were categorized as *very extreme*. Additionally, *n* = 24 participated in a relay team of 4, and *n* = 60 in a team of 6. Those athletes were categorized as being *less extreme*. Information on training specific data and how they performed in the race are listed in Table 1. Prior to participation, athletes were asked to fill out and sign an informed consent form. The procedure was fully in line with the guidelines laid out in the Declaration of Helsinki and was not subject to a separate IRB assessment per the guidelines of the ethics committee of the University Konstanz.

### 2.2. Race Setting

The 24 h race took place on the 12 and 13 June 2021 in Reichenau, Germany. The course led over a 3.5 km loop, which should be covered as often as possible within 24 h as a single starter or in a relay team. Running was allowed to be paused at any time to rest or to change to another team member. For the break, an assigned camping area was available to all participants. In addition to their own food, drinks and bars were permanently provided by the organizer. The course was not fully illuminated at night, which made it necessary to run with a head or body light. The best three teams in each category were awarded with non-monetary prizes at the end of the competition.

### 2.3. Questionnaire

Demographics, sport-specific trait boredom, self-regulatory challenges, and potential action crises during the race were assessed with two surveys that had to be completed before (version 1: expectation) and after the race (version 2: actual experience). Not all surveyed questions are analyzed and reported in the present paper, but the complete questionnaire can be viewed (https://osf.io/2tj94/?view_only=b60a598024f6469a8eb2ae90e57443aa). The response rate of usable tests of version 1 was 55% (109 returned questionnaires) and 40% for version 2 (80 returned questionnaires). An overview of the questions asked and the number of responses that were obtained is provided in the flow diagram in Figure 1.

#### 2.3.1. Expectation: Before the Race

To capture sports-specific trait boredom, eight questions were adapted from the boredom subscale of the Achievement Emotions Questionnaire (AEQ) [44]. This approach of adapting parts of the AEQ to the conditions of interest (here: a running competition) has been successfully performed before in the domain of sports and exercise [5]. Additionally, previous research has provided evidence for criterion, convergent, and discriminant validity of such a sport-specific adaptation of the AEQ’s boredom subscale [5]. Furthermore, McDonald’s Omega of the scale was 0.802 (CI 95%: 0.72–0.86), indicating good reliability of the scale. To tailor for the event, items and the questionnaire description were adjusted so that they applied to the sport of running. Items had to be answered on a 5-point Likert-type scale (1 = strongly agree–5 = strongly disagree). Example items were “Running bores me to death” or “I am bored”.

Then, athletes were queried for the degree to which they expected to be facing specific self-regulatory challenges during the run. This was surveyed by a 10-point Likert-type scale (1 = strongly agree–10 = strongly disagree) with a question for boredom (“I will get very bored during my run.”), willpower (“I will use a lot of willpower during my run.”), effort (“I will exert a lot of physical efforts during my run.”), and pain (“I will experience a lot of physical pain during my run.”).

Finally, the demographic data age, gender, as well as the start category and average running kilometers per week were assessed.

#### 2.3.2. Actual Experience: After the Race

After the race, participants answered four questions that queried the degree to which the four self-regulatory challenges mattered during the competition. Again, this was performed via single item measures for boredom (“I was very bored during my run.”), willpower (“I applied a lot of willpower during my run.”), effort (“I exerted a lot of physical efforts during my run.”), and pain (“I suffered a lot of physical pain during my run.”), and the questions had to be answered on the same 10-point Likert scale.

To assess, if an action crisis had occurred, athletes reported if they had encountered a crisis during the competition. For this, a 0–24 h timeline was provided to participants, and they indicated timepoints at which they had encountered a crisis. This measure was then dichotomized into a yes/no format for further analyses. If athletes reported to have had one or more crisis, this was categorized as “yes”, and if no crisis was reported, this was categorized as “no”. Furthermore, they were asked what distance (in kilometers) they covered over the 24 h period.

### 2.4. Statistical Analysis

To test the hypothesis that ultra-endurance athletes display lower sport-specific boredom scores than their less extreme peers, a one-sided independent sample *t*-test with the mean boredom score as the dependent variable was conducted. To assess, if experienced boredom, pain, willpower, and effort differed during the run, a repeated measures ANOVA was conducted. Start-type was added as a covariate to check if the outcome is independent of the groups (*very* vs. *less extreme starters*). If the repeated measures ANOVA yielded a significant *F-*value, this was followed up by post-hoc tests, to assess specific differences between the self-regulatory challenges. To test if the self-regulatory challenges were present at all (i.e., if they differed from them minimum point on the scale), a series of one-sample *t*-tests was conducted. To assess if the degree to which each of the four challenges was experienced would predict if athletes would face a crisis during the race, we conducted a multiple logistic regression. Here, crisis (yes/no) was used as the dependent variable, and the four challenges (boredom, pain, effort, willpower) were included as predictors. All operations were performed and Figure 2, Figure 3, Figure 4 and Figure 5 were created using JASP (version 0.14.1).

## 3. Results

### 3.1. Descriptive Statistics

Table 1 shows the descriptive statistics for the 24 h race participants. Very extreme, compared to the less extreme athletes, reported a significantly higher weekly training volume (*t*(97) = 6.5, *p* < 0.001, *d* = 1.6) and also covered a greater distance during the race (*t*(73) = 6.8, *p* < 0.001, *d* = 2.1). This attests to the validity of our categorization in *very extreme* and *less extreme* athletes. Highlighting the importance of crises during competition, 51.36% of the athletes stated that they had experienced at least one crisis during the event.

### 3.2. Comparing Sport-Specific Trait Boredom between More or Less Extreme Athletes

Concerning sport-specific trait boredom, the statistical analysis revealed a significant difference in favor of a lower sport-specific trait boredom in the *very extreme* runners, compared to their *less extreme* peers, *t*(102) = −2.1, *p* = 0.024, d=−0.48 (Table 1 and Figure 2) (As assumptions for parametric testing were violated, this was followed up by non-parametric tests, which did not meaningfully alter the results, Mann-Whitney U = 1202.50, *p* = 0.008). Thus, although both groups scored boredom values at the lower end of the scales range, these were even lower in those who chose to compete as single starters or duo starters.

### 3.3. Boredom during An Ultra-Endurance Competition

Next, we assessed the degree to which athletes reported having experienced self-regulatory challenges during the competition. As displayed in the raincloud plot (Figure 3), the degree to which challenges were experienced varied substantially across the types of challenges, *F*(3, 237) = 242.0, *p* < 0.001, η2=0.75). Boredom was by far the lowest (1.64 ± 1.1), willpower and effort had the highest scores (7.6 ± 2.6 and 8.3 ± 2.1), and pain was in the middle range (6.0 ± 2.8). Post hoc tests revealed significant differences between all challenges (boredom vs. pain (*p* < 0.001, d=−1.8); vs. effort (*p* < 0.001, d=−2.7); vs. willpower (*p* < 0.001, d=−2.5); pain vs. effort (*p* < 0.001, d=−0.9); vs. willpower (*p* < 0.001, d=−0.7); effort vs. willpower (*p* = 0.02, d=0.3)). This result still remains, even when the start type is included as a covariate. Additional, one-sample t-tests revealed that all challenges were different from 1, and that all *p’s* < 0.001. This indicates that none of the surveyed challenges was completely irrelevant during the race.

To assess how sport-specific trait boredom and expected as well as experienced challenges covaried, Pearson correlation analyses were conducted (Figure 4). Attesting to the criterion validity of the sport-specific trait boredom scale, these scores were significantly positively correlated with expecting to struggle with boredom, as well as with actually struggling with boredom. In addition, willpower, pain, and effort rather consistently shared strong positive correlations with each other (both with respect to expecting to struggle with them, and with respect to actually struggling with them). Expecting to suffer from boredom was linked to actually struggling with boredom, but not with other variables. A slightly different pattern emerged when looking at how experienced boredom covaried with the other challenges. Here, a significant positive correlation with experienced pain emerged, whereas correlations with willpower (*p* = 0.053) and effort (*p* = 0.110) were not significant but the descriptive results were in the same direction.

The logistic regression analysis that assessed if the experienced self-regulatory challenges predicted the occurrence of a crisis was significant, Χ^2^(75) = 22.9, *p* < 0.001). Interestingly, only boredom was a significant predictor (*p* = 0.02), whereas neither willpower, effort, nor pain were significant predictors for facing a crisis. In contrast to a balanced chance relationship for the challenges of willpower, effort, and pain (odds ratios (OR) = 1.2, 1.0, 1.1), when boredom was rated higher, the probability of having also experienced a crisis increased (OR = 12.5). Emphasizing the robustness of this finding, this did not meaningfully change after we controlled for age, gender, level of extremeness, and weekly training kilometers. This indicates that although boredom is experienced far less than the other challenges, it is the only challenge we surveyed that predicted the experience of a crisis during the competition. Figure 5 highlights that athletes who reported boredom levels of three (or greater) were all but certain to encounter a crisis during the run.

## 4. Discussion

In the current study, we investigated boredom among ultra-endurance athletes. Consistent with our expectations, we found that ultra-endurance athletes reported lower sport-specific boredom when compared to less extreme runners. In addition, compared to the more prototypical competition-induced challenges, such as pain, willpower, and effort, ultra-endurance athletes reported boredom at a much lower intensity during a 24 h running competition. Crucially, although boredom was reported less intensely compared to the other challenges, athletes nevertheless reported boredom levels that were different from the value of 1, which would have indicated that they did not experience boredom at all. This highlights that boredom is a challenge during sports, even among athletes who self-select themselves into this sport. Most importantly, boredom mattered with respect to whether or not athletes were facing a crisis during their competition. Only boredom—and neither pain, effort, nor willpower—was a significant predictor of encountering a crisis during competition.

By investigating boredom in the context of ultra-endurance, this research addresses a hitherto unaddressed gap and, in what follows, we will discuss two aspects of our results in more detail. First, our findings showed that ultra-endurance runners and less extreme endurance athletes differed in how much they experienced boredom. This is consistent with the idea that boredom plays a role with regard to whether or not people engage in sports or not. Second, compared to more prototypical endurance exercise-induced challenges, boredom was much less pronounced during a competition. However, it was the only one of the surveyed challenges that was a predictor of experiencing a crisis during the competition. This indicates that competitors might be good at coping with self-regulatory challenges that are typical for them, as opposed to when struggling with those challenges that are less intensely experienced.

### 4.1. Ultra-Endurance Athletes and Less Extreme Athletes Differ in Sport-Specific Boredom

The present study was conducted to provide insights into the challenges of ultrarunners, including state and trait boredom, which has so far been neglected by research on endurance sports. We found that the most extreme ultra-endurance runners (those who compete in a 24 h running competition as a single starter or in relay teams of twos) associated running less with boredom compared to endurance athletes who were less extreme (runners who competed in the same event in relay teams of four or six). At first sight, it might be somewhat trivial to conclude that the more people are bored by sport, the less they engaged in it. However, given that physical inactivity is a public health pandemic [45], and inactivity rates in Western countries remain largely unchanged [46], it remains crucial to better understand what deters so many people from regular exercise. On this matter, the relevance of boredom has long been overlooked. The present findings extend previous findings that found a negative relationship between sport-specific trait boredom and physical exercise [5,47]. Thus, even within our sample of highly active athletes, differences in sport-specific trait boredom differentiated those who partook in the most demanding endurance challenge from those who did not. This further substantiates the idea that sport-specific boredom is relevant in the sports and exercise setting.

One explanation for the lower tendency of ultra-endurance athletes to associate their sport with boredom could be their engagement in regular training (Table 1) and their competition experience. Regular and consistent training poses a self-regulatory challenge that needs to be mastered if one wants to become a good athlete. The very extreme athletes’ high weekly training load might indicate that these athletes are very good at self-regulation (at least when it comes to training for their sports). Self-regulation and trait boredom share a robust negative correlation [5,48]. Thus, it is conceivable that good athletes display an adaptive self-regulatory profile with regard to shielding goal striving (i.e., becoming a good athlete) from external and internal distractions [5]. This would mean that fairly low boredom prone individuals would be more likely to commit to endurance sports. At the same time, in the light of the nature vs. nurture debate [49], it would be conceivable that the sport and its demands has shaped the athletes towards better self-regulation as a consequence of the athletic involvement (see [50] for a discussion in the context of team sports). Given the possibility that sports participation can aid the development of an adaptive self-regulatory profile, it would be an interesting research question to assess if doing sports can lower sport-specific trait boredom, and if this might even generalize to boredom proneness in general.

Better self-regulatory skills might also allow good athletes to better plan their activity (e.g., tactical arrangement) in such a way that the chosen level of difficulty best matches their capabilities [6]. Indeed, research shows that good runners have less variance in their self-pacing than worse runners [51]. Again, this indicates that athletes might self-regulate more effectively [31] and, therefore, are less likely to experience boredom during sports. This is also in line with research on flow (a state that is decidedly different from boredom), indicating that athletes can differ in their ability and frequency to exercise in a state of flow [52], which makes goal striving easier. Thus, experienced athletes might be well equipped to choose athletic challenges and to engage with them in a fashion that minimizes boredom, thereby making it easier and more valuable for them to be physically active.

### 4.2. Being Bored Is A Recipe for Tumbling into Crises

Secondly, the roles of pain, effort, willpower, and boredom were evaluated as potential self-regulatory challenges during the 24 h race. It was observed that, as expected from the literature [30,31], high levels of physical effort and pain were experienced, and willpower had to be exerted. It turns out that ultra-endurance athletes very rarely get bored by a 24 h long effort, which from the outside appears to be highly monotone. This highlights the fact that even very monotonous (in the eyes of a general person) activities must not be boring per se, and that athletes might be extremely robust against the sensation of boredom. Crucially, though, even very low levels of boredom appeared to matter greatly. Only boredom was a significant predictor of encountering a crisis during the competition. This effect was robust, even when we controlled for variables, such as age, gender, weekly training volume, and level of extremeness (i.e., whether or not they took part as single starters or ran in a team relay). Thus, although ultra-endurance athletes seem to be very robust against experiencing boredom, they appear to be not very well versed in dealing with boredom when it occurs.

The present study offers some first insights into the relevance of boredom in ultra-endurance sports. From an applied perspective, it seems to be instructive to adopt a proactive approach for dealing with boredom during a competition. Athletes and coaches tend to focus a lot on dealing with challenges, such as willpower or pain, for example, by practicing mental training [53], or, more negatively tainted, by taking painkillers to stop experiencing aches and pain [54]. As these challenges were present to a very high degree in our sample, this focus certainly seems to be a sensible one. Given that those challenges were not predictive of encountering a crisis, it is conceivable that athletes are already very proficient at dealing with those demands. However, if even low levels of boredom are related to experiencing a crisis during competition, such structured approaches to dealing with boredom seem to be called for too. This is a topic that has so far not received much research interest. However, cases of ultra-endurance athletes who actively practice dealing with boredom for upcoming challenges [55], and the use of music or podcasts as distractions in endurance sports [56], highlight the relevance of preparing for boredom during competition and training. Certainly, to design and test effective “boredom trainings”, further applied research is needed to better understand boredom’s role during training and during competition.

## 5. Limitations

Lastly, while the present study helps to address an important gap in psychological research on the relevance of boredom in ultra-endurance athletes, some limitations should be addressed. First, we cannot ascertain that a crisis was caused by boredom. Even if it was the only statistically relevant predictor, other unobserved factors might be at play. One such factor might be fatigue, as research outside of the sports setting has shown that boring situations might increase fatigue [14]. However, while it is very plausible that boredom and fatigue covary during an ultra-endurance competition, it appears rather unlikely that such an unaccounted-for variable would only cause a spurious relationship between boredom and having a crisis, but not with the other challenges which appear to be much more proximally related to physiological parameters of resource depletion.

In addition, the temporal dynamics of boredom over the course of a 24 h challenge are most likely not linear. Research outside the sporting context has highlighted the need to take the dynamics of boredom into account [57]. To account for this, future research could employ a higher temporal resolution in tracking boredom during the course of an ultra-endurance competition. For example, this could be performed by surveying athletes’ sensations on every completed lap/hour. This would also offset the potential for event-related sensations that are reported after the race being biased if they are assessed after the event.

Lastly, boredom is an everyday sensation, and empirical evidence showed that self-reports do a good job at capturing meaningful correlates and consequences of boredom [4]. Consistent with this, in our study, the trait measure was robustly related with whether or not athletes got bored during competition. Still, for future research it might be instructive to dig deeper into the mechanisms by which boredom alters behavior in field research. A promising method might be to include more downstream questions about feelings or situational perceptions that help to elucidate the consequences of boredom more directly. For example, asking whether the athlete would have, phase wise, preferred to pursue an activity other than running could indirectly elicit a statement about boredom’s consequences.

## 6. Conclusions

Extreme athletes are less likely to associate their sport with boredom than less extreme athletes. In addition, the experience of boredom, unlike prototypical competition-induced challenges, such as pain, willpower, and effort, is a significant predictor of encountering a crisis during a race. This emphasizes that athletes and coaches might want to consider incorporating “boredom training” into the context of race-specific preparations, as well as making sure that the sport an athlete ends up specializing in should ideally be not boring to the athlete. Future work, with a higher temporal acquisition of challenge data during an event, will further specify the relevance and practical implementation of the current findings.

## Figures and Tables

**Figure 1 ijerph-19-06859-f001:**
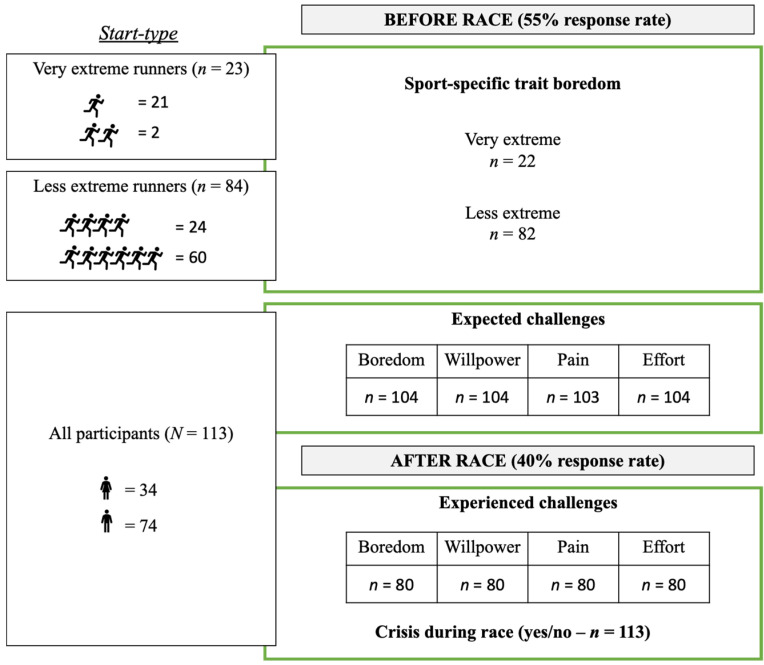
The different questionnaires or single items were not fully answered by all participants. This overview shows transparently which quantity of responses were included in the statistics. The visualization shows, analogous to the applied statistical methods, data for the question about trait boredom, a separation into very extreme and less extreme, as well as for all other questions answered by the whole sample.

**Figure 2 ijerph-19-06859-f002:**
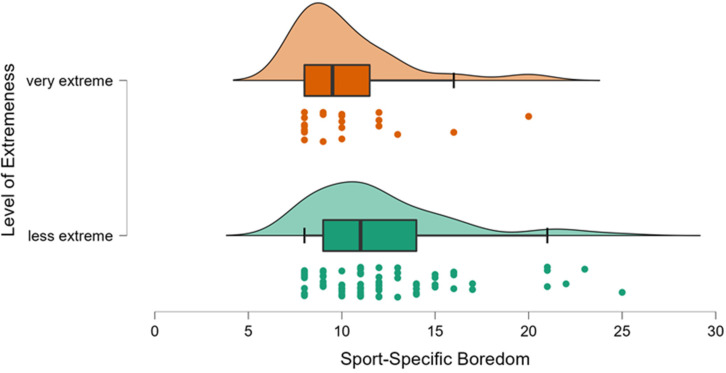
Raincloud plots comparing the ultra-endurance athletes (denoted as “very extreme”) with their “less extreme” peers.

**Figure 3 ijerph-19-06859-f003:**
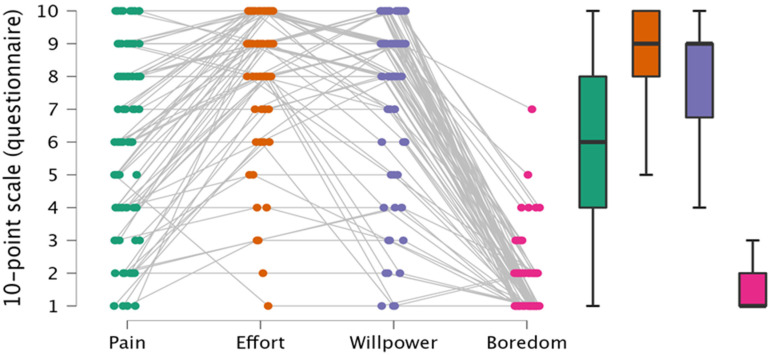
Post-race (experienced) rating of all participants concerning relevant challenges.

**Figure 4 ijerph-19-06859-f004:**
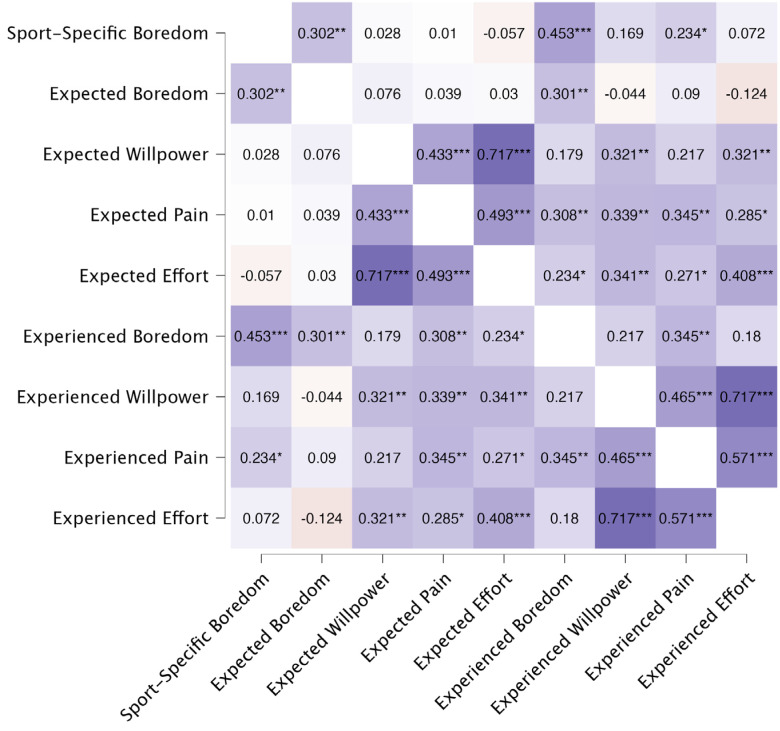
Heatmap reporting correlations (coefficient. r in the according box) between all challenges and the sport-specific boredom score. Significant cases are flagged with asterisks (* *p* < 0.05, ** *p* < 0.01, *** *p* < 0.001).

**Figure 5 ijerph-19-06859-f005:**
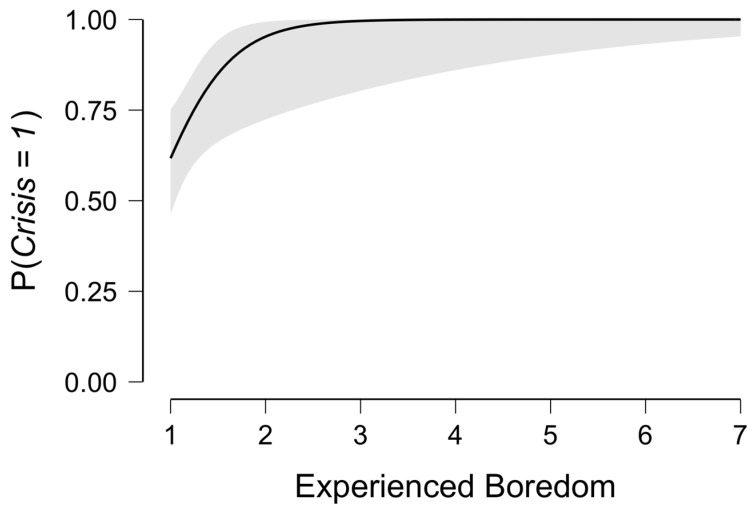
Boredom and probability of experiencing a crisis.

**Table 1 ijerph-19-06859-t001:** Participation, age, and training, as well as event performance statistics.

	Very Extreme Athletes					
	**N**	**Age**	**Weekly Training (km)**	**Avg. Event Distance (km)**	**Max. Event Distance (km)**	**Sport-Specific Boredom (Avg. Mean Score)**
male	18	44.4 ± 14.9	72.8 ± 33.6	110.7 ± 55.7	197	1.27 ± 0.37
female	5	37.0 ± 9.4	60.0 ± 39.6	82.5 ± 57.3	123	1.32 ± 0.43
all	23	43.9 ± 13.7	72.6 ± 34.0	104.8 ± 54.4	197	1.28 ± 0.37
	**Less Extreme Athletes**					
	**N**	**Age**	**Weekly Training ** **(km)**	**Avg. Event Distance (km)**	**Max. Event Distance (km)**	**Sport-Specific Boredom (Avg. Mean Score)**
male	54	38.5 ± 13.3	37.5 ± 21.4	54.9 ± 14.3	101	1.54 ± 0.53
female	30	31.8 ± 12.4	33.6 ± 15.5	43.8 ± 13.2	80	1.46 ± 0.35
all	84	36.0 ± 13.3	35.3 ± 19.6	50.7 ± 14.8	101	1.50 ± 0.46

## Data Availability

Original data and further supplementary materials can be accessed online: https://osf.io/2tj94/?view_only=b60a598024f6469a8eb2ae90e57443aa. Correspondence concerning this article should be addressed to the corresponding author.

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
