# Peer review of "24 Hours on the Run—Does Boredom Matter for Ultra-Endurance Athletes’ Crises?"

_ijerph, 2022, doi:10.3390/ijerph19116859_

Round 1

Reviewer 1 Report

The article constitutes an interesting and sound research paper. It is well-written, well-structured, and captivating. I don't have any comments but two extremely insignificant ones.

1) the caption of the Figure 2 is misplaced, instead of being just below the figure is two lines below;

2) the role of fatigue is overlooked - boredom may be prevented by fatigue associated with an extreme effort - this can also explain the lower boredom rates among more extreme athletes.

Reviewer 2 Report

The manuscript is well written and addressed methodological concerns (e.g., the classification of athletes as  ultra or less extreme, how crisis is defined).  A few grammatical mistakes need to be addressed ( line 37, spacing).

Author Response

The manuscript is well written and addressed methodological concerns (e.g., the classification of athletes as  ultra or less extreme, how crisis is defined).  A few grammatical mistakes need to be addressed ( line 37, spacing).

Response: We would like to thank you for your support and the positive feedback on our work. We have revised the paper again grammatically and linguistically.

Reviewer 3 Report

The manuscript is well written and the presentation of the results is well adequate.

However, it is necessary to make significant changes to the questionnaire used.

Does it have a reliability or kappa index?
Has it been used in previous research?
Does it have any kind of validity?

Please provide more information regarding the main study instrument.

Reviewer 4 Report

This paper is very interesting. Thank you for the opportunity to review this interesting article. The study is innovative and addresses important information on the area of sport and exercise . Even though, the manuscripts present some flaws that must be considered. I recommend its publication after minor changes. Also, I noted a small number of places where a few words should be changed in the English language use.

1.Please describe the chapter research participants in more detail.

2. The general objective and specific objectives should appear at the end of the introduction. The objective should be clearly written, referring to the population, the intervention, the comparison and the results (PICO strategy).

3. The criteria for the inclusion and exclusion of participants must be indicated.

4. Control of participant losses must be specified through a flowchart.
